# Multi-omics insights into host-viral response and pathogenesis in Crimean-Congo hemorrhagic fever viruses for novel therapeutic target

**Ujjwal Neogi[1,2]\*[†], Nazif Elaldi[3†], Sofia Appelberg[4], Anoop Ambikan[1], Emma Kennedy[5,6], Stuart Dowall[5], Binnur K Bagci[7], Soham Gupta[1], Jimmy E Rodriguez[8], Sara Svensson-Akusjärvi[1], Vanessa Monteil[9], Akos Vegvari[8], Rui Benfeitas[10], Akhil Banerjea[11], Friedemann Weber[12], Roger Hewson[5,6,13], Ali Mirazimi[4,9,14]\***

[1]The Systems Virology Lab, Division of Clinical Microbiology, Department of Laboratory Medicine, Karolinska Institute, ANA Futura, Campus Flemingsberg, Stockholm, Sweden; [2]Manipal Institute of Virology (MIV), Manipal Academy of Higher Education, Manipal, India; [3]Department of Infectious Diseases and Clinical Microbiology, Medical Faculty, Cumhuriyet University, Sivas, Turkey; [4]Public Health Agency of Sweden, Solna, Sweden; [5]Public Health England, Porton Down, Salisbury, United Kingdom; [6]Oxford Brookes University, Oxford, United Kingdom; [7]Department of Nutrition and Dietetics, Faculty of Health Sciences, Sivas Cumhuriyet University, Sivas, Turkey; [8]Division of Chemistry I, Department of Medical Biochemistry and Biophysics, Karolinska Institutet, Stockholm, Sweden; [9]Division of Clinical Microbiology, Department of Laboratory Medicine, Karolinska Institute, ANA Futura, Campus Flemingsberg, Stockholm, Sweden; [10]National Bioinformatics Infrastructure Sweden (NBIS), Science for Life Laboratory, Department of Biochemistry and Biophysics, Stockholm University, Stockholm, Sweden; [11]National Institute of Immunology, Aruna Asaf Ali Marg, New Delhi, India; [12]Institute for Virology, FB10-Veterinary Medicine, Justus-Liebig University, Giessen, Germany; [13]Faculty of Infectious and Tropical Diseases, London School of Hygiene and Tropical Medicine, London, United Kingdom; [14]National Veterinary Institute, Uppsala, Sweden

**\*For correspondence:**
ujjwal.neogi@ki.se (UN);
ali.mirazimi@
folkhalsomyndigheten.se (AM)

[†]These authors contributed equally to this work

**Competing interest:** The authors declare that no competing interests exist.

**Abstract** The pathogenesis and host-viral interactions of the Crimean–Congo hemorrhagic fever orthonairovirus (CCHFV) are convoluted and not well evaluated. Application of the multi-omics system biology approaches, including biological network analysis in elucidating the complex host-viral response, interrogates the viral pathogenesis. The present study aimed to fingerprint the system-level alterations during acute CCHFV-infection and the cellular immune responses during productive CCHFV-replication in vitro. We used system-wide network-based system biology analysis of peripheral blood mononuclear cells (PBMCs) from a longitudinal cohort of CCHF patients during the acute phase of infection and after one year of recovery (convalescent phase) followed by untargeted quantitative proteomics analysis of the most permissive CCHFV-infected Huh7 and SW13 cells. In the RNAseq analysis of the PBMCs, comparing the acute and convalescent-phase, we observed system-level host's metabolic reprogramming towards central carbon and energy metabolism (CCEM) with distinct upregulation of oxidative phosphorylation (OXPHOS) during CCHFV-infection. Upon application of network-based system biology methods, negative coordination of the biological signaling systems like FOXO/Notch axis and Akt/mTOR/HIF-1 signaling with

metabolic pathways during CCHFV-infection were observed. The temporal quantitative proteomics in Huh7 showed a dynamic change in the CCEM over time and concordant with the cross-sectional proteomics in SW13 cells. By blocking the two key CCEM pathways, glycolysis and glutaminolysis, viral replication was inhibited in vitro. Activation of key interferon stimulating genes during infection suggested the role of type I and II interferon-mediated antiviral mechanisms both at the system level and during progressive replication.

## Editor's evaluation

The data presented here provide novel insight into the host response to CCHFV infection. These data further our understanding of how CCHFV causes disease in humans and will support the development of therapeutics to address the significant morbidity and mortality caused by this virus.

## Introduction

Crimean–Congo hemorrhagic fever orthonairovirus (CCHFV), a negative-sense RNA virus belonging to the *Nairoviridae* family, is a major emerging pathogen with an increasing number of outbreaks all over the world. Causing a mild-to-severe viral hemorrhagic fever (CCHF; Crimean–Congo hemorrhagic fever) poses a substantial threat to public health due to its high mortality rate in humans (3–40%), modes of transmission (tick-to-human/animal, animal-to-human, and human-to-human) and geographical distribution. CCHF is endemic in almost 30 countries in sub-Saharan Africa, South-Eastern Europe, the Middle East, and Central Asia (*Bente et al., 2013*; *Zivcec et al., 2016*). The ixodid ticks, especially those of the genus *Hyalomma*, are both a vector and a reservoir for CCHFV and are highly ubiquitous with their presence in more than 40 countries (*Gargili et al., 2017*). In recent years, CCHFV outbreaks have become more frequent and expanded to new geographical areas. This has been attributed to climate change and the spread of infected ticks by birds and the livestock trade. The presence of the CCHFV tick vector in Portugal, Spain, Germany, and even Sweden (*Grandi et al., 2020*) and England (*McGinley et al., 2021*) highlights the need for stricter surveillance due to the possibility of a future intrusion (*Estrada-Peña et al., 2012*). Turkey has reported the highest number of laboratory-confirmed CCHF cases and is one of the worst affected countries in the world (*Monsalve-Arteaga et al., 2020*). Since the first identification in 2002 up till the end of 2019, a total of 11,780 confirmed CCHF cases have been reported with a case-fatality rate of 4.7% (unpublished data by the Turkish Ministry of Health). There were nearly 500 cases every year, reported mainly during the summer months May-July (*Ak et al., 2020*).

Because of the sporadic nature of CCHF outbreaks in humans in the endemic regions, a lack of infrastructure, and the absence of systematic studies, little is known about the pathogenesis and host-virus interactions during the acute phase of CCHF disease and associated sequelae after recovery. An in-depth understanding of host responses to CCHFV is necessary to design better therapeutic and containment strategies for CCHF. The systems biology studies using -omics approaches on patient material and infected cells can elucidate potential host immune response mechanisms and disease pathogenesis. Application of the multi-omics system biological methods can also distinguish disease severity as reported recently in 16 viruses, including severe acute respiratory syndrome coronavirus 2 (SARS-CoV-2), Chikungunya, Zika, Ebola, Influenza viruses (*Appelberg et al., 2020*; *Krishnan et al., 2021*; *Zheng et al., 2021*). However, no studies elucidating the host viral response using advanced system biological methods were reported for CCHFV infection.

Here, we have applied global blood transcriptomics in longitudinal samples collected during the acute phase of CCHFV-infection and the convalescent phase (nearly after a year of recovery) to measure the system-wide changes during the CCHFV-infection in patients from Turkey. We also performed temporal quantitative proteomics analysis to understand the cellular alterations during the productive CCHFV-infection in two different cell lines, human adrenal carcinoma cell line, SW13, and human liver cell line Huh7 that were reported to be the most permissive cell lines for CCHFV (*Dai et al., 2021*). Using the newly gained insights, we then modulated the critical pathways by drugs to halt the productive CCHFV-replication in in vitro infection models. Our study thus provides a comprehensive, system-level picture of the regulation of cellular and metabolic pathways during productive CCHFV-infection that can aid in identifying novel therapeutic targets and strategies.

**eLife digest** Crimean-Congo hemorrhagic fever (CCHF) is an emerging disease that is increasingly spreading to new populations. The condition is now endemic in almost 30 countries in sub-Saharan Africa, South-Eastern Europe, the Middle East and Central Asia. CCHF is caused by a tick-borne virus and can cause uncontrolled bleeding. It has a mortality rate of up to 40%, and there are currently no vaccines or effective treatments available.

All viruses depend entirely on their hosts for reproduction, and they achieve this through hijacking the molecular machinery of the cells they infect. However, little is known about how the CCHF virus does this and how the cells respond. To understand more about the relationship between the cell's metabolism and viral replication, Neogi, Elaldi et al. studied immune cells taken from patients during an infection and one year later.

The gene activity of the cells showed that the virus prefers to hijack processes known as central carbon and energy metabolism. These are the main regulator of the cellular energy supply and the production of essential chemicals. By using cancer drugs to block these key pathways, Neogi, Elaldi et al. could reduce the viral reproduction in laboratory cells.

These findings provide a clearer understanding of how the CCHF virus replicates inside human cells. By interfering with these processes, researchers could develop new antiviral strategies to treat the disease. One of the cancer drugs tested in cells, 2-DG, has been approved for emergency use against COVID-19 in some countries. Neogi, Elaldi et al. are now studying this further in animals with the hope of reaching clinical trials in the future.

## Results

### Samples and clinical data

In this study, 18 samples were collected during the acute phase of the disease with a median time of 4 days (range 1–6 days) after the onset of symptoms. We used the severity grading scores (SGS) to define the CCHF severity that calculated using age, clinical findings (bleeding, hepatomegaly, organ failure), routine laboratory parameters (blood levels of liver enzymes and lactate dehydrogenase, blood platelet, and leucocyte counts, blood coagulation tests [prothrombin time, D-dimer and fibrinogen]; *Bakir et al., 2012*). By using these criteria, a standard SGS sheet for each patient was filled by the infectious diseases physician on admission day. By using SGS criteria, 33% (6/18) patients were grouped into severity group 1 (SG-1), 61% (11/18) patients into severity group 2 (SG-2), and, 6% (1/18) patients into severity group 3 (SG-3). The median age of the patients was 49 years (range: 18–79), and 12 (66.7%) of the patients were male. A 79-year-old male patient in SG-3 died on the third day of hospitalization. The case-fatality rate (CFR) for the cohort was 5.6%. Follow-up samples were collected from 12 individuals after a median duration of 54 weeks (range: 46–57 weeks). The CCHF patient characteristics are summarized individually in *Table 1* and the calculated daily SGS scores during hospitalization in *Table 1—source data 1*.

### System-level metabolic reprogramming during the acute phase of CCHFV infection

Due to the natural heterogeneity in human cohorts, we used longitudinal samples from 12 patients (SG-1: n = 5; SG-2: n = 7) to perform differential expression analyses for each infected patient between the time of infection and approx. 1 year post-recovery (Range: 46–57 weeks). The differential gene expression (DGE) profile for the acute phase compared to the recovered phase in all patients showed an upregulation of 2891 genes and a downregulation of 2738 genes (adj. p<0.05)(*Figure 1A* and *Supplementary file 1*). To check whether the gene expression changes between the acute phase and recovered phase may be due to differences in cell types abundances, we performed digital cell quantification (DCQ) using the Estimating the Proportions of Immune and Cancer cells (EPIC) (*Racle and Gfeller, 2020*) algorithm for blood circulating immune cells. No statistically significant (adj p < 0.05) difference was observed in the key immune cell types (*Figure 1—figure supplement 1*). Next, we used the functional analysis using a consensus scoring approach based on multiple gene set analysis (GSA) runs by incorporating the directionality of gene abundance using R/Bioconductor package

**Table 1.** The CCHF patient characteristics.

| PID | Age | Gender | The date of symptoms onset | The date of hospitalization | Time to hospitalization (days) | The date of the first sampling | The date of the second sampling | SGS score | Severity group** | Rt-pcr | CT-values | Anti-CCHFV IgM | Outcome |
|---|---|---|---|---|---|---|---|---|---|---|---|---|---|
| P01 | 33 | Female | 30 May 2017 | 03 June 2017 | 4 | 03 June 2017 | 05 July 2018 | 5 | 1 | Positive | 31,85 | ND | Survived |
| P02 | 18 | Male | 06 June 2017 | 12 June 2017 | 6 | 12 June 2017 | ND | 7 | 2 | Positive | 25,89 | positive | Survived |
| P03 | 45 | Male | 12 June 2017 | 13 June 2017 | 1 | 14 June 2017 | 01 July 2018 | 0 | 1 | Positive | 21,87 | ND | Survived |
| P04 | 67 | Male | 13 June 2017 | 16 June 2017 | 3 | 17 June 2017 | 05 July 2018 | 8 | 2 | Positive | 22,38 | ND | Survived |
| P05 | 48 | Male | 12 June 2017 | 18 June 2017 | 6 | 19 June 2017 | 08 July 2018 | 7 | 2 | Positive | 29,79 | ND | Survived |
| P06 | 68 | Male | 13 June 2017 | 19 June 2017 | 6 | 20 June 2017 | 05 July 2018 | 5 | 1 | Positive | 28,41 | ND | Survived |
| P07 | 77 | Male | 19 June 2017 | 22 June 2017 | 3 | 23 June 2017 | 05 July 2018 | 6 | 2 | Positive | 24,77 | ND | Survived |
| P08 | 29 | Female | 20 June 2017 | 24 June 2017 | 4 | 25 June 2017 | 02 July 2018 | 6 | 2 | Positive | 26,91 | ND | Survived |
| P09 | 50 | Female | 20 June 2017 | 25 June 2017 | 5 | 26 June 2017 | 06 July 2018 | 4 | 1 | Positive | 26,36 | Positive | Survived |
| P10 | 35 | Female | 07 July 2017 | 12 July 2017 | 5 | 12 July 2017 | 04 July 2018 | 3 | 1 | negative | NA | Positive | Survived |
| P11 | 64 | Female | 15 July 2017 | 18 July 2017 | 3 | 19 July 2017 | ND | 10 | 2 | Positive | 22,46 | ND | Survived |
| P12 | 57 | Male | 16 July 2017 | 21 July 2017 | 5 | 22 July 2017 | 09 July 2018 | 9 | 2 | Positive | 20,81 | ND | Survived |
| P13 | 79 | Male | 22 July 2017 | 24 July 2017 | 2 | 24 July 2017 | ND | 11 | 3 | Positive | 22 | ND | Died |
| P14 | 36 | Male | 01 August 2017 | 06 August 2017 | 5 | 07 August 2017 | 04 July 2018 | 7 | 2 | Positive | 24,66 | ND | Survived |
| P15 | 62 | Male | 15 August 2017 | 20 August 2017 | 5 | 21 August 2017 | 06 July 2018 | 9 | 2 | Positive | 19,86 | ND | Survived |
| P16 | 48 | Male | 05 September 2017 | 07 September 2017 | 2 | 07 September 2017 | ND | 4 | 1 | Positive | 22,09 | ND | Survived |
| P17 | 55 | Male | 12 April 2018 | 17 April 2018 | 5 | 18 April 2018 | ND | 9 | 2 | Positive | 26,16 | ND | Survived |
| P18 | 44 | Female | 23 April 2018 | 27 April 2018 | 4 | 29 April 2018 | ND | 9 | 2 | Positive | 21,27 | ND | Survived |

ND: not determined; NA: not applicable; SGS: severity grading system; RT-PCR: real-time - polymerase chain reaction; CT: cycle threshold; CCHFV: Crimean-Congo hemorrhagic fever virus.

*1: Low (0–5); 2: Intermediate (6-10); 3: High (11-16).

The online version of this article includes the following source data for table 1:

**Source data 1.** Severity grade scoring during hospitalization.

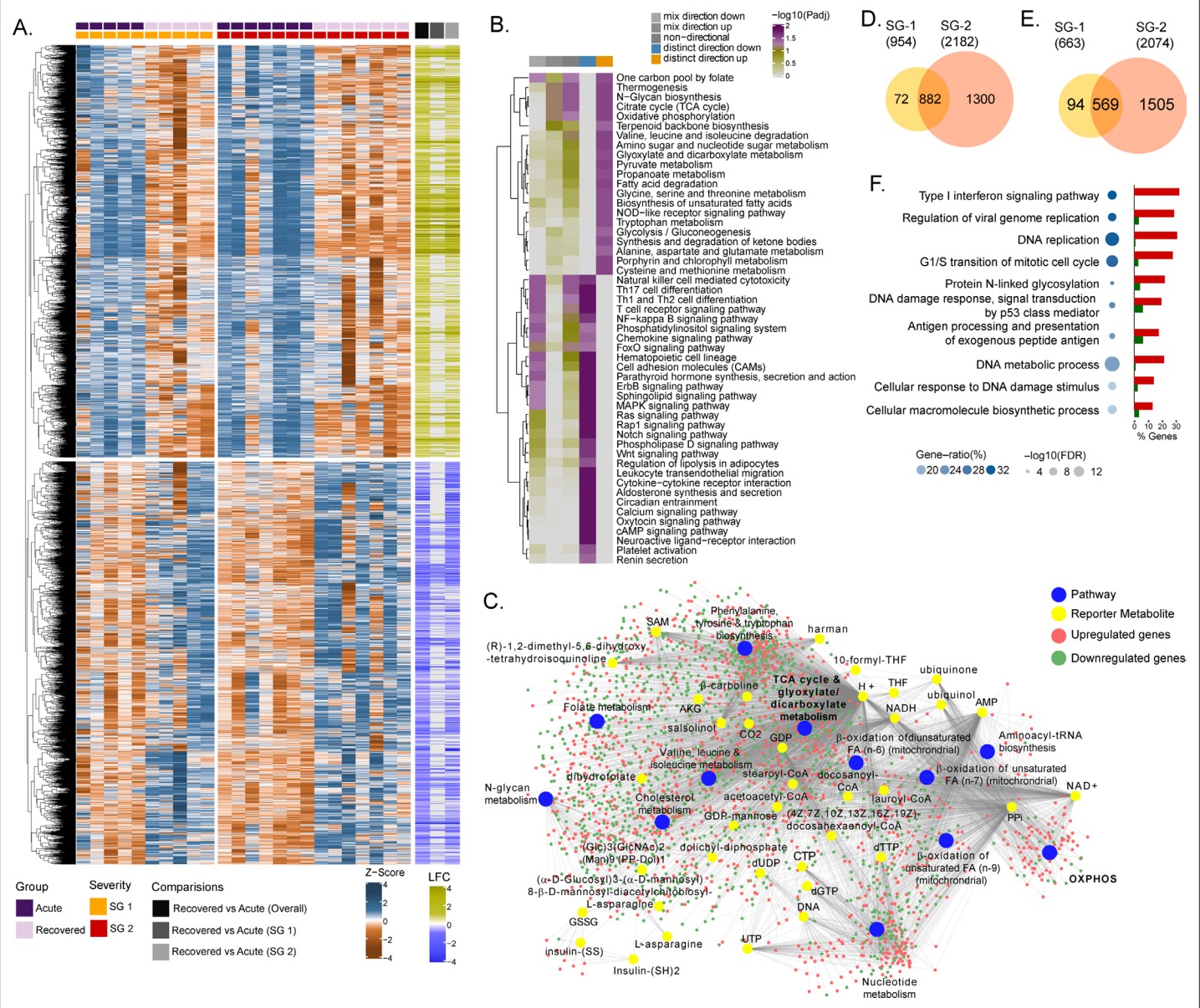

**Figure 1.** Differential gene expression and pathway analysis between acute and recovery phases. (**A**) Heatmap of Z-score transformed expression values of significantly regulated genes in the pair-wise comparisons namely recovered vs. acute (overall), recovered vs. acute (SG-1), recovered vs. acute (SG-2). The columns represent the patient samples and their corresponding severity groups at different time points. The rows represent genes that are hierarchically clustered based on Euclidean distance. (**B**) Pathways were found to be significantly regulated (adj. p < 0.05) by genes expressed at the acute infection phase compared to recovered phase. The heatmap visualizes negative log scaled adjusted p-values of different directionality classes. Non-directional p-values were generated based on gene-level statistics alone without considering the expression direction. The mixed-directional p-values were calculated using subset of gene-level statistics of up and down-regulated genes respectively for mixed-directional up and down. Distinct directional up and distinct directional down p-values are calculated from gene statistics with expression direction (**C**) Network visualization of significant reporter metabolites (adj. p < 0.1) and reporter subsystems (pathways) identified in acute compared to recovered. The yellow node denotes reporter metabolite and blue node denotes reporter subsystems. Light red and green colored nodes represent upregulated and downregulated genes respectively. Each edge in the network denotes association of genes with reporter metabolites and subsystems based on the human genome-scale metabolic model. (**D**) Venn diagram of significantly up-regulated genes in recovered vs acute (SG-1) and recovered vs acute (SG-2) phases (**E**) Venn diagram of significantly down-regulated genes in recovered vs. acute (SG-1) and recovered vs. acute (SG-2) phases. (**F**) Gene ontology (GO, biological process) enrichment analysis results of commonly regulated genes (882 upregulated and 569 down-regulated) from (**D**) and (**E**). The color gradient and bubble size correspond to the gene ratio of each GO term and the adjusted p-value of the enrichment test, respectively. The adjacent bar graph represents the percentage of genes upregulated or downregulated in each GO term.

The online version of this article includes the following figure supplement(s) for figure 1:

*Figure 1 continued on next page*

*Figure 1 continued*

**Figure supplement 1.** Digital cell quantification using EPIC.

**Figure supplement 2.** Severity group association with gene expression.

**Figure supplement 3.** Violin plot of 22 soluble markers as determined from Luminex assay assays.

PIANO (*Väremo et al., 2013*) for KEGG pathway gene-set. Using the group-specific consensus scores (acute vs. recovered) and directionality classes, we identified distinct upregulation (adj. p < 0.05) of metabolic pathways such as one carbon pool by folate, oxidative phosphorylation (OXPHOS), glycolysis, N-glycan biosynthesis, and antiviral pathways like the NOD-like receptor signaling pathway (*Figure 1B* and *Supplementary file 2*). However, the pathways related to the down-regulated genes were mainly antiviral defense mechanism-associated pathways including innate immune responses like Th1, Th2, and Th17 cell differentiation, the NF-kB pathways, chemokine signaling pathway, etc. (*Figure 1B*). Additionally, since most of the metabolic pathways were upregulated, we used the DGE results of acute-vs-recovered to identify reporter metabolites. Reporter metabolites are metabolites around which most of the transcriptional changes occur (*Patil and Nielsen, 2005*) thus being indicative of gene-level altered regulation of metabolism. The analysis identified 37 significantly upregulated reporter metabolites (adj. p < 0.1), that were part of OXPHOS, TCA-cycle, nucleotide metabolism, N-glycan metabolism, and amino acid-related pathways (*Figure 1C*). To specifically investigate the genes that were significantly associated with disease severity during the acute phase, the samples were grouped into either SG-1 or SG-2 and 3 combined. There were 12 genes (*ERG, PROM1, HP, HBD, AHSP, CTSG, PPARG, TIMP4, SMIM10, RNASE1, VSIG4, CMBL, MT1G*) that were significantly upregulated in patients in the SG-2 and SG-3 combination group compared to SG-1 (*Figure 1—figure supplement 2A*) However, no obvious links between these genes were noted and no apparent clustering was observed (*Figure 1—figure supplement 2B*). This was further supported by serum secretome analysis using the 22 soluble cytokine and chemokine markers by Luminex assay on samples collected during the acute phase of the disease from SG-1 (n = 6) and SG-2 (n = 11). Of the 22 markers used for analysis, only interleukin 8 (IL-8) and Granulocyte-macrophage colony-stimulating factor (GM-CSF) was shown borderline significance between SG-1 and SG-2 (*Figure 1—figure supplement 3*). However, when we compared the acute phase with the recovered phase in SG-1, and SG-2 separately, there was a distinct DGE profile. In SG-1 the differentially expressed genes were significantly fewer (adj. p < 0.05; n = 1617, upregulated: 954 and downregulated: 663) compared to those in SG-2 (adj. p < 0.05; n = 4256, upregulated: 2182 and downregulated: 2074) (*Figure 1D and E*). There were 1451 overlapping genes between SG-1 and SG-2 that were differentially upregulated (n = 882) and downregulated (n = 569). Using gene ontology (GO) analysis after removal of the redundant terms using REVIGO (*Supek et al., 2011*), the majority of the genes from the top two GO terms that were significantly upregulated were part of the IFN-I signaling pathway (GO:0060337) and the regulation of viral genome replication (GO:0045069) (*Figure 1F*). This indicates that the disease severity significantly affected gene expression of the interferon signaling pathway profiling during the acute phase, whereas it was comparable when they recovered.

## Distinct interferon signaling-related pathways in CCHFV-infection

To identify the CCHFV-induced changes in the interferon-related signaling pathways, we used our previously curated datasets for genes (n = 205) associated with the interferon response (*Chen et al., 2021*). The majority of the genes of the interferon signaling pathways were upregulated (36%, 73/205, adj p < 0.05), while 11% (22/205) were downregulated (*Figure 2A*). Of the IFN-regulated genes, IFI27 (ISG12) showed the most robust upregulation (*Figure 2B*). This was further supported by RNAscope analysis targeting the IFI27 transcript in the SW13 cell line infected with CCHFV strain IbAr10200 (*Figure 2C*). Apart from this ISG20, ISG15, Mx1, Mx2, and several other ISGs showed upregulation in the acute phase (*Supplementary file 1*). Given that the interferon signaling pathways have a role in disease severity, we next performed an association between the patient viral load and genes in the interferon signaling-related pathways. We identified six genes (*TRIM25, IFI35, EIF2AK2, USP18, IFI6, and BST2*) that were negatively associated with the cycle threshold (CT) value of RT-PCR (adj p < 0.05 and R > −0.8, *Figure 2D*), suggesting a higher viral load was associated with an increased expression of these ISGs. Overall, the gene expression data indicate that the CCHFV infection regulates IFN

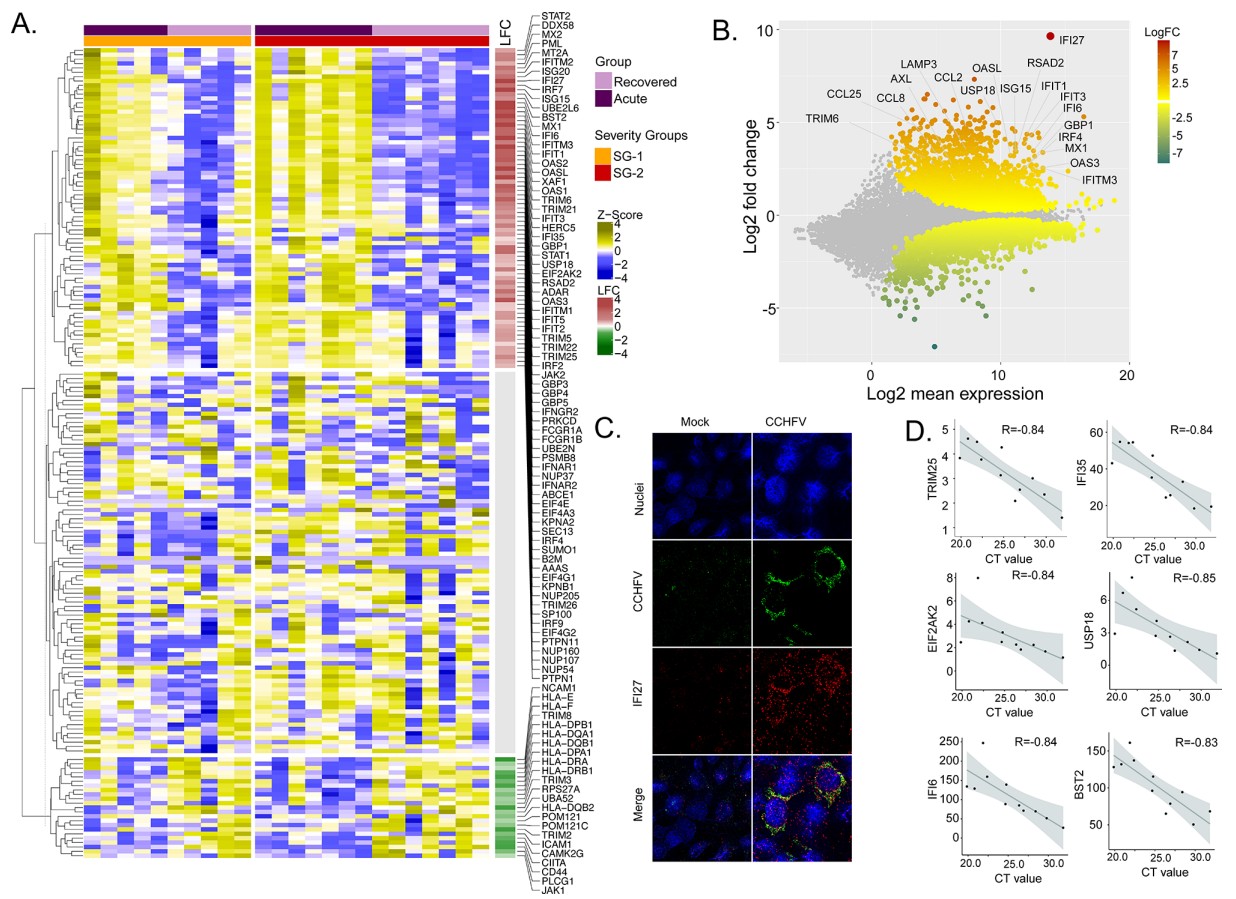

**Figure 2.** Differentially expressed genes in interferon (IFN) signaling pathways. (**A**) Heatmap visualizes the expression pattern of IFN-signaling genes (including ISGs) significantly different between the recovered and acute phases. The columns represent the patient samples and their corresponding severity groups at different time points. The rows represent genes hierarchically clustered based on Euclidean distance. (**B**) MA-plot of differentially regulated genes between the recovered and acute phases. ISGs are marked. (**C**) RNAscope analysis targeting IFI27 genes in infected and non-infected cells. (**D**) Spearman correlation between viral load and IFN signaling genes (adj p < 0.05).

responses and patients with a successful disease outcome showed stimulation of several ISGs during the acute phase of infection.

## Network analysis identified the central role of central carbon and energy metabolism (CCEM) in the regulation of signaling pathways

To further deepen our understanding of the cellular regulation of acute CCHFV-infection at the molecular level from a systems perspective, we employed a weighted gene co-expression network analysis at the transcriptomic level. Based on the network analysis of pairwise gene co-expression (adj. p < 0.001, Spearman $\rho$ > 0.84), we identified a set of seven communities of strongly interconnected genes (*Figure 3A*). Next, we ranked all the communities based on their centrality (average degree of nodes) to identify the sets of genes with the highest coordinated expression changes that were predicted to influence network behavior robustly. The functional enrichment analysis of the central community (c1) of the transcriptomics is associated (adj. p < 0.05) mainly with alterations in pyruvate metabolism, TCA-cycle, and to a smaller extent to glycolysis and gluconeogenesis (adj. p < 0.2) (*Figure 3A* and *Figure 3—figure supplement 1*). Further, we observed (*Figure 3B* and *Figure 3—figure supplement 2*) a high number of negative correlations between community (c1) and those associated with Notch, mechanistic target of rapamycin (mTOR) and Forkhead box protein O (FoxO) signaling (c5), and hypoxia inducing factor-1 (HIF-1) signaling (c7). Interestingly, the OXPHOS-associated community (c3) also tends to be negatively correlated with those involved in Notch/mTOR/FoxO signaling (c5) and HIF-1 signaling (c7). These patterns are also observed among the top 10% of most central genes

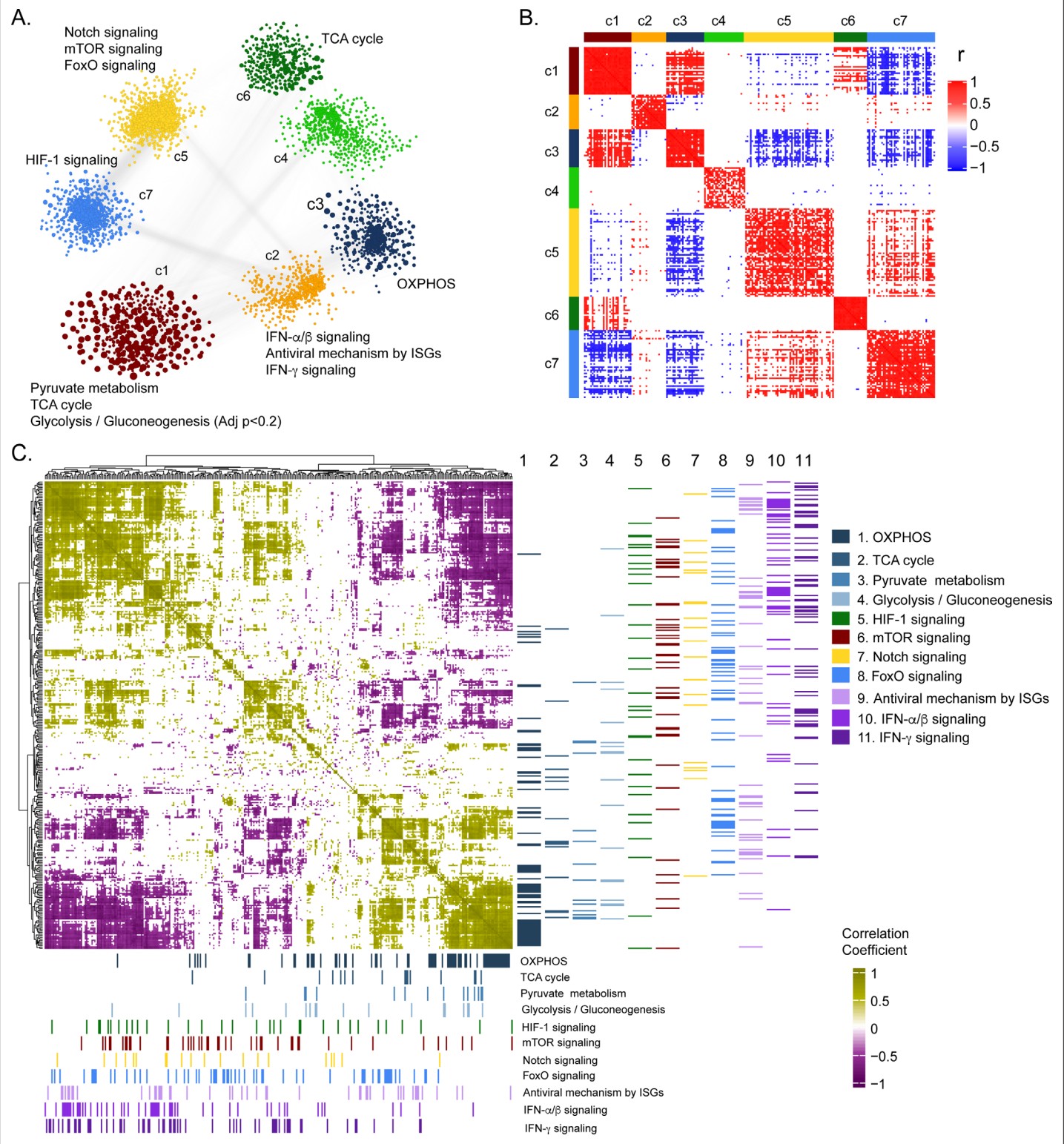

**Figure 3.** Weighted co-expression network analysis. (**A**) Network visualization of seven gene co-expression communities identified. Nodes and node size represent genes and their centrality (degree) respectively and edges represent significant Spearman correlation (adj p < 0.001 and $R > 84$). Key significantly regulated pathways (adj. p < 0.05) in each community are labeled. (**B**) Heatmap of correlations among top 5% central genes in each community. Column and row annotation denotes corresponding communities. (**C**) Heatmap of significant correlation (adj. p < 0.05) between key metabolic and signaling pathways mentioned in (**A**). Column and row annotation denotes corresponding pathways.

The online version of this article includes the following figure supplement(s) for figure 3:

*Figure 3 continued on next page*

*Figure 3 continued*

**Figure supplement 1.** Gene set enrichment analysis of the individual communities.

**Figure supplement 2.** Weight co-expression network of the negatively co-related genes We observed a high number of negative correlations between this community (**c1**) and those associated with Notch, mTOR, and FoxO signaling (**c5**) and HIF-1 signaling (**c7**).

in each community, suggesting key opposite differences not only at a global community level but also in key genes in each community (*Figure 3B*). At a pathway level, we indeed observed antagonistic trends between the above-mentioned pathways (*Figure 3C*). Our functional and network community analyses in the patient transcriptomics identified the coordination of biological signaling systems like FoxO, Notch and mTOR/HIF-1 signaling with metabolic pathways of CCEM during CCHFV-infection.

## Quantitative proteomics analysis identified modulation of key metabolic processes and signaling pathways during productive replication in vitro

Our longitudinal transcriptomics analysis of CCHF patient samples revealed alternations in the several key metabolic processes and signaling pathways during the acute phase of infection at a system level. As CCHFV fails to infect the peripheral blood mononuclear cells (PBMCs) (*Connolly-Andersen et al., 2009*), to understand the global changes in the cellular response during productive CCHFV-infection, we infected Huh7 and SW13 cells with CCHFV, which are the common cell lines used in pathogenesis studies and considered highly permissive for CCHFV (*Dai et al., 2021*). To allow multiple rounds of infection we used a multiplicity of infection (MOI) of 1 and used a time-course proteomic experiment for 24 and 48hpi using single batch TMT-labeling based mass-spectrometric analysis to avoid batch effects, inflated false-positive results, and minimize the typical missing values issue (*Brenes et al., 2019*). Due to the higher cell death the proteomics analysis could not be performed in SW13 48hpi. In the UMAP clustering of the proteome data, we observed a clear separation between the mock and virus-infected cells in both the cell lines (*Figure 4A*). At 24hpi and 48hpi a substantial amount of CCHFV proteins, N, M, and L protein were detected (*Figure 4B*). The immune fluorescence analysis targeting N-protein of CCHFV infected Huh7 cells at 24hpi with 1 MOI is shown in *Figure 4C*. The differential protein analysis (DPA) identified 3205 and 3070 proteins upregulated and 2926 and 3279 proteins downregulated in the infected samples at 24hpi and 48hpi in Huh7 cells and 2,217 upregulated and 1705 downregulated in SW13 cells respectively compared to the mock (adj. p < 0.05) (*Supplementary file 3*). The consensus scoring-based gene set analysis (GSA) using PIANO on the DPA at 24hpi and 48hpi in Huh7 and 24hpi in SW13 identified 68 pathways to be dysregulated in at least one of the comparisons. We observed downregulation (adj. p < 0.05) of the glycolysis/gluconeogenesis, purine metabolism, PI3K-Akt, and HIF-1 signaling pathways in both Huh7 and SW13 cell lines at 24hpi (*Figure 4D*) indicating CCHFV utilized these pathways during productive replication at an early phase. These pathways are known to have feedback mechanisms (*Hayward, 2004*; *Locasale, 2018*) to maintain cellular homeostasis, which is consistent with the observation that at 48hpi (in Huh7 cells) the pathways were not significantly dysregulated. The pathways like TCA-cycle and insulin secretion showed opposite trends in the cell lines indicating cell type-specific differential regulation of metabolic and signaling pathways during CCHFV replication. In time-series analysis in Huh7 cells, oxidative phosphorylation (OXPHOS) pathway was upregulated during CCHFV infection in a temporal manner indicating shift in metabolic processes towards OXPHOS during productive replication of the virus. The other pathways that also showed distinct temporal upregulation during CCHFV infection in vitro were N-glycan biosynthesis and cytokine-cytokine receptor interactions. In turn, pathways like FoxO signaling, T-cell receptor signaling pathways, Th1 and Th2 cell differentiation, and NK cell-mediated cytotoxicity were downregulated and upregulated of Notch signaling in Huh7 24hpi but not at 48hpi indicating the role of these pathways at the early stage of infection. A severe metabolic rearrangement occurred in SW13 cells at 24hpi toward central carbon and energy metabolism and amino acid metabolism as the pathways like pyruvate metabolism, glycine, serine, and threonine metabolism, tryptophan metabolism etc. were downregulated (*Figure 4D*). We also performed quantitative proteomics analysis of the Huh7 cells with 4 MOI infections at 24hpi and observed similar alterations in the pathways (*Figure 4—figure supplement 1*). Next, we performed gene set enrichment analysis (GSEA ) in Enrichr and compared Huh7 and SW13, 24hpi and patients RNAseq data and observed the

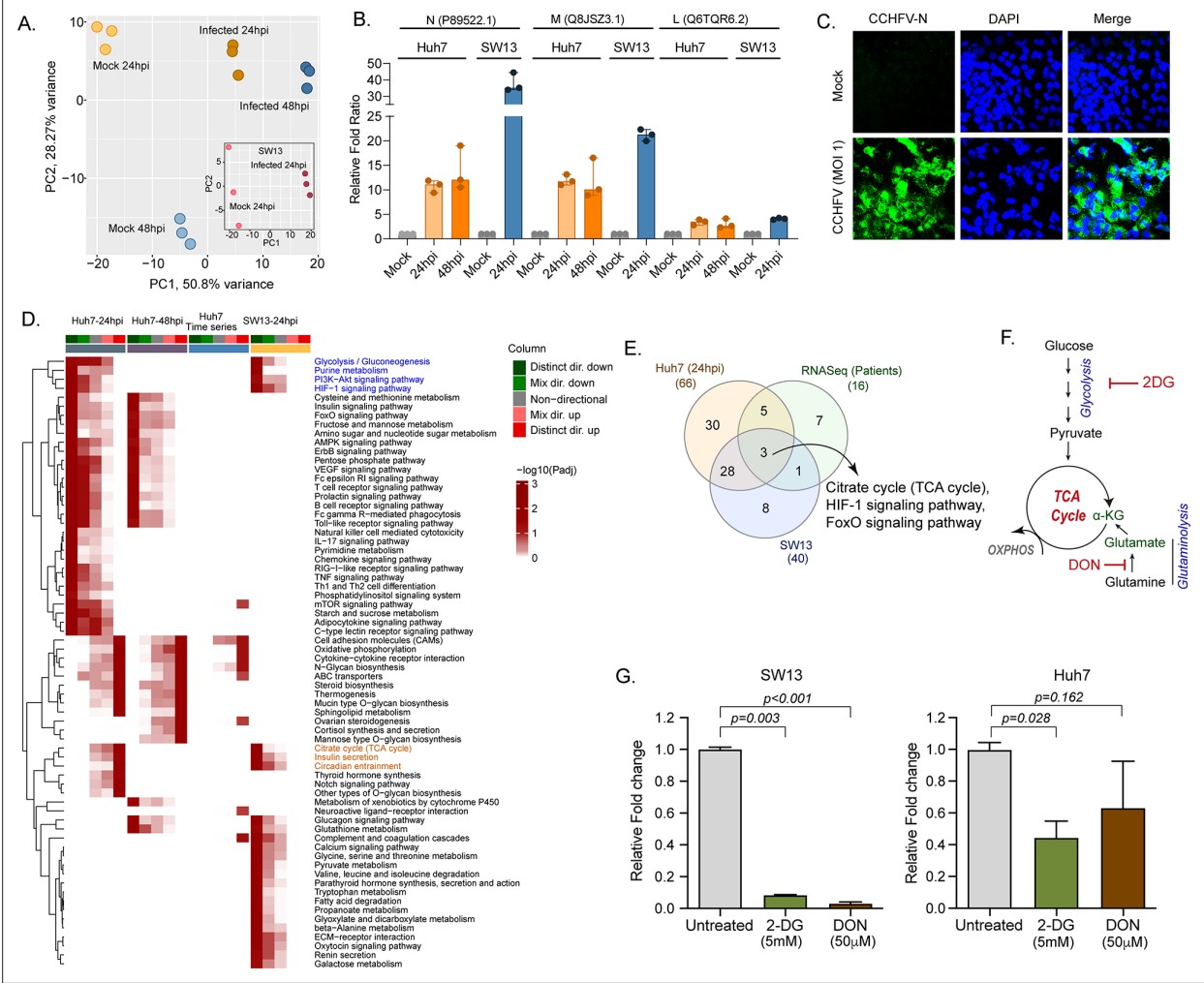

**Figure 4.** LC-MS/MS-based quantitative proteomics analysis in CCHFV-infected Huh7 and SW13 cells. (**A**) Principal component analysis of proteomics samples of Huh7 cells and SW13 (inset) using only human proteins. (**B**) Identification of the CCHFV N (UniProtKB P89522.1), M (UniProtKB Q8JSZ3.1) and L (UniProtKB Q6TQR6.2) protein in the quantitative proteomics analysis. (**C**) Immunofluorescence staining of the CCHFV nucleoprotein to assess the infectivity. (**D**) Significantly regulated pathways (adj p < 0.05) in any of the pair-wise proteomics analyses in Huh7 and SW13 cells. The heatmap visualizes negative log scaled adjusted p-values of different directionality classes. Non-directional p-values are generated based on gene-level statistics alone without considering the expression direction. The mixed-directional p-values are calculated using subset of gene-level statistics of up and down-regulated genes respectively for mixed-directional up and down. Distinct directional up and distinct directional down p-values are calculated from gene statistics with expression direction. The first column annotation represents directionality of pathways and second column annotation denotes corresponding differential expression analysis. (**E**) Venn diagram showing commonly dysregulated pathways in patients transcriptomics and cell line proteomics. (**F**) Schematic diagram of the glycolysis and glutaminolysis and targeted drugs. (**G**) Metabolic control of viral replication in vitro. Fold change of the CCHFV *L-gene* following infection and treatment of 2-DG and DON at indicated concentrations compared to untreated in SW13 cells and Huh7 cells. A two-tailed paired Student *t*-test was performed, and p values are mentioned.

The online version of this article includes the following figure supplement(s) for figure 4:

**Figure supplement 1.** Quantitative proteomics of the Huh7 with 4 MOI infection 24hpi and comparisons with the 1 MOI infection indicated 2452 proteins were common that were significantly dysregulated.

key common dysregulated pathways were TCA cycle, HIF-1, and FoxO signaling pathways (*Figure 4E*). Glycolysis and OXPHOS are molecular interconversion systems, where the end product of the glycolysis is fueling OXPHOS through the TCA cycle which normally is the primary energy source and major pathways of CCEM. Glutaminolysis is an alternative pathway for mitochondrial energy production through OXPHOS under altered metabolic conditions (*Zhang et al., 2019*; *Zheng, 2012*). Therefore, we blocked glycolysis and glutaminolysis in SW13 and Huh7 cells using 2-deoxy-D-glucose (2-DG) (5 mM) and 6-diazo-5-oxo-L-norleucine (DON) (50 µm), respectively (*Figure 4F*) following infection.

Infectivity of CCHFV, quantified as relative CCHFV *L-gene* levels in cells lysates, showed a significant decrease in 2-DG treated cells in both SW13 and Huh7 (p = 0.003 and p = 0.028, respectively). While in the DON treated cells a significant decrease was observed in SW13 cell (p < 0.001) and an inhibitory trend in Huh7 (p = 0.162) (*Figure 4G*). These data indicate that alteration in the CCEM affects CCHFV replication despite the cell-specific differences.

## Temporal dynamics of interferon response in vitro

The temporal changes in the interferome (cluster of interferon genes) are represented as a heat-map in *Figure 5A* and the log2fold change of the significantly altered protein levels at 24hpi and 48hpi are represented as volcano plots in *Figure 5B*. Several ISGs, such as Mx1, Mx2 IFIT1, ISG15, ISG20, and IFI6, were transcriptionally upregulated in the acute phase in patient samples (Supplementary Data File 1), were also significantly elevated in proteomics of infected Huh7 cells by 48hpi (*Figure 5C*). To determine that the observed induction of ISGs is due to the CCHFV-infection itself and not caused by the presence of any residual interferon in the virus-containing supernatant, we performed infection using UV-inactivated virus supernatant. As shown in the immunoblots in *Figure 5D*, a significant increase in expression of several ISGs namely RIG-I, IFIT1, ISG15 and a noticeable increase in Mx1, Mx2, and ISG20 proteins were observed in CCHFV-infected cells and not in UV inactivated virus supernatant, confirming that CCHFV-infection induces the expression of these ISGs. The WB images from all three experiments were given in *Figure 5—figure supplement 1* and the *Figure 5—source data 1*.

## Discussion

In our study, using the system level genome-wide transcriptomic analysis of a longitudinal patient cohort, temporal quantitative proteomics from in vitro infection assays in Huh7 cells, cross-sectional quantitative proteomics analysis in SW13 cells, and in vitro inhibition of CCHFV replication following the blocking the glycolysis and glutaminolysis, we showed that during CCHFV-infection there is metabolic reprogramming of host cells towards central carbon and energy metabolism and this plays a major role in viral replication despite the existence of cell-type-specific differences. Upregulation of OXPHOS was a unique feature of CCHFV-infection, at both the system-level blood transcriptomics and cellular proteomics during productive infection in Huh7. By applying network-based system biology methods, we identified the negative co-ordination of the biological signaling systems like FoxO/Notch axis and mTOR/HIF-1 signaling along with metabolic pathways of CCEM during CCHFV-infection at the system level. Blocking the two key CCEM pathways, glycolysis and glutaminolysis, controlled viral replication in vitro. Moreover, IFN-I mediated antiviral mechanisms were also activated with elevated key antiviral ISGs (ISG12, ISG15, ISG20), and MXs (Mx1 and Mx2).

Viruses exploit the host metabolic machinery to meet their biosynthetic demands. This reliance is further highlighted by observed variations in the cell-specific viral replications and production leading to changes in host metabolism (*Yu et al., 2011*). The changes in the energy metabolism can therefore be seen as an evolving property of the combined host-virus metabolic system and could be related to changes in host cellular demands arising from viral production (*Molenaar et al., 2009*). Our system-level transcriptomics data on patient material and in vitro cell culture assays indicated a transient dysregulation of key metabolic processes of the CCEM, like OXPHOS, glycolysis, and TCA-cycle in CCHFV-infection. These pathways are also known to promote replication of several other RNA viruses including human immunodeficiency virus type 1 (HIV-1), rubella virus, dengue virus (DENV), rhinovirus, hepatitis C virus (HCV), influenza virus, etc (*Mayer et al., 2019*; *Thaker et al., 2019*). Blocking glycolysis and glutaminolysis, that fuel OXPHOS, resulted in severe suppression of CCHFV-replication suggesting the need for these pathways for efficient viral replication. Our system biology analysis further indicated the coordinating role of the metabolic pathways of CCEM with biological signaling systems like Notch/FoxO axis and mTOR/HIF-1 signaling during the CCHFV-infection. It is known that these biological systems regulate energy metabolism. Notch signaling plays an essential role in maintaining the cellular energy homeostasis via regulation of HIF-1 and PI3K/AKT signaling that is known to induce glycolysis (*Landor et al., 2011*).

On the other hand, FoxO signaling regulates cell proliferation by modulating energy metabolism and gluconeogenesis (*Kousteni, 2012*). The coordinated role of these transcriptional regulators (HIF-1α, FoxO, mTOR, and Notch1) modulates OXPHOS and mitochondrial biogenesis (*Kondo et al.,*

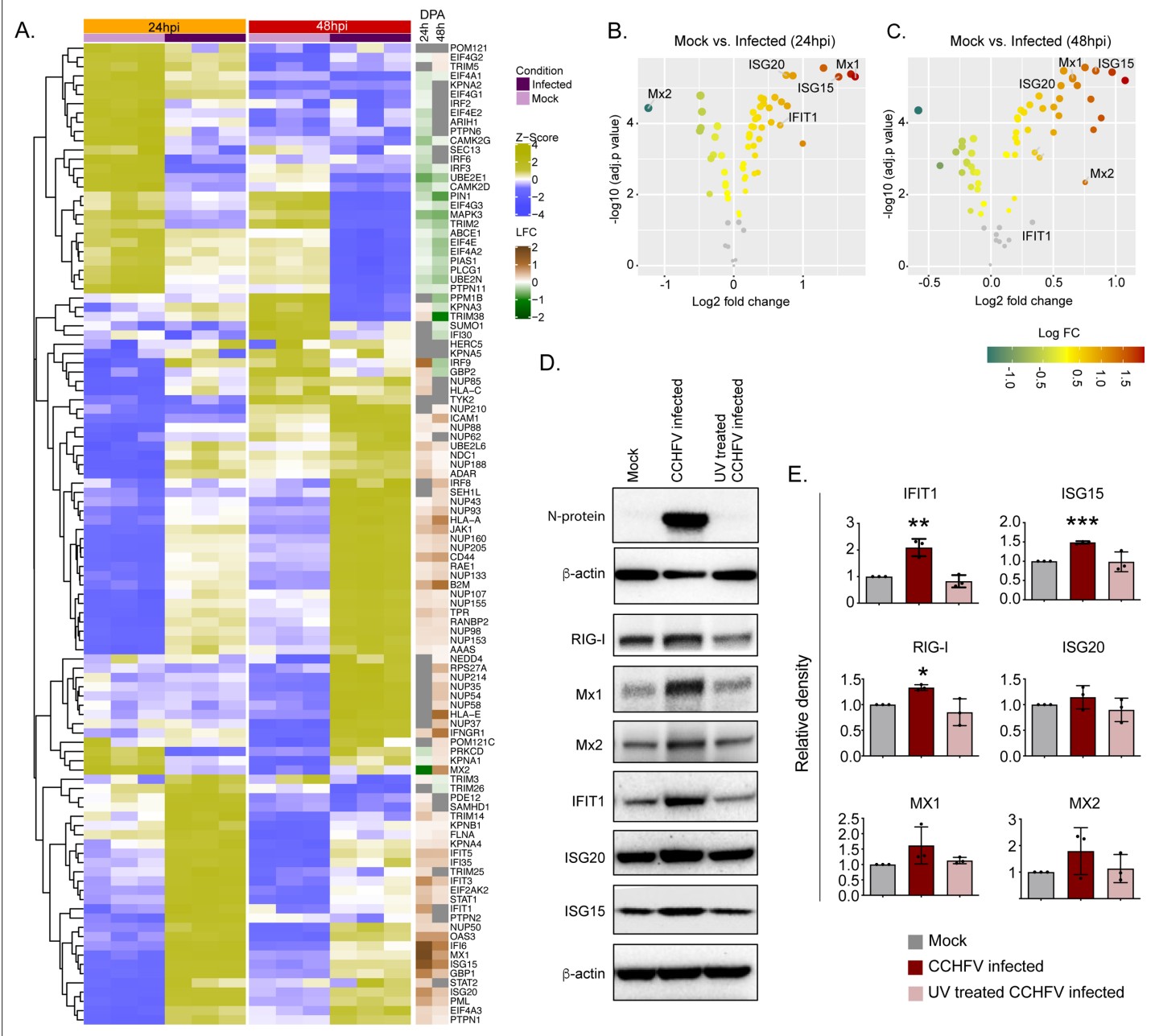

**Figure 5.** Temporal dynamics of interferon stimulating genes (ISGs). (**A**) Heatmap of Z-score transformed expression values of proteins belonging to the cellular response to IFN signaling pathways in Mock-infected and CCHFV-infected Huh7 cells at 24hpi and 48hpi as identified in proteomics. The log-2-fold change in the genes corresponding to the indicated proteins identified in our patient transcriptomics data (recovered vs acute) is shown under the column name RNASeq. (**B and C**) Volcano plot of ISGs visualizing the expression status of Mock-infected and CCHFV-Infected samples at (**B**) 24hpi and (**C**) 48hpi. The size and color gradients of the dots correspond to the adjusted P values of differential expression analysis and the log2 fold change, respectively. (**D**) Representative western blots illustrate the indicated ISGs in Mock-infected, CCHFV-infected, and UV-inactivated CCHFV-infected Huh7 cells at 48hpi. ISG20 antibody gave a specific band at approx. 40 kDa without any non-specific band in the membrane that was cut at 50 kDa in the top. (**E**) The densitometric intensity of the bands was quantified using Fiji (ImageJ) software. The intensity of the individual bands was first normalized to the respective β-actin loading control and further relative normalization with respect to the mock-infected control was done. The bars are represented as means ± SD of three independent experiments. A two-tailed paired Student *t*-test was performed, and p values are represented as *p < 0.05, **p < 0.01 and ***p < 0.001.

The online version of this article includes the following source data and figure supplement(s) for figure 5:

**Source data 1.** Raw western blot images.

**Figure supplement 1.** Western blot Images of ISGs (RIG-I, IFIT1, Mx1, Mx2, ISG20, ISG15), CCHFV-N protein and β-actin at 48hpi from three experimental replicates.

*2020*). Notch signaling has also been known to facilitate viral infectivity of RNA viruses including influenza virus, respiratory syncytial virus (RSV), HCV, etc. (*Breikaa and Lilly, 2021*) and have regulatory roles in inflammation (*Shang et al., 2016*). Our study is concordant with an earlier study that reported the downregulation of the Notch signaling in CCHFV-infection at the transcript level (*Arslan et al., 2019*). However, our study also pointed out that during productive infection in Huh7 cells, Notch signaling was upregulated at 24hpi but not at 48hpi, indicating a role in the early stage of viral replication. Silencing of the Notch1 reported increasing toll-like receptor 4 (TLR4) triggered proinflammatory cytokines (*Zhang et al., 2012*) which is common during acute CCHFV-infection (*Ergönül et al., 2017*).

Several viruses encode proteins such as Ebola virus (EBOV) glycoprotein, the Dengue virus (DENV) nonstructural protein 1 (NS1), etc., that are known to activate TLR4 (*Olejnik et al., 2018*). On the other hand, at the first encounter with the pathogen, PI3Ks negatively regulate TLRs including TLR4 signaling (*Fukao and Koyasu, 2003*). Of note, in our proteomics data during productive infection, we observed downregulation of PI3K/Akt signaling at 24hpi but not at 48hpi. lthough there was no distinct downregulation of the whole pathway, in our patient system-level transcriptomics data, we also noted significant downregulation of genes belonging to the PI3K/Akt pathway during acute CCHFV-infection. Moreover, apart from PI3K/Akt, mTOR and HIF-1 signaling were also downregulated at 24hpi, indicating modulation of PI3K/mTOR/HIF-1 axis by CCHFV for its replication. In our previous study, we have shown that exogenous nitric oxide that is known to regulate the HIF-1 via the Akt/mTOR pathway under normoxic conditions (*Sandau et al., 2000*), inhibited CCHFV in vitro (*Simon et al., 2006*). Interestingly, an in vitro study in another *Bunyavirus*, Rift Valley fever virus (RVFV), identified the inhibition of the PI3K/Akt pathway by dephosphorylation of the AKT and Forkhead box protein O1 (FoxO1)(*Popova et al., 2010*). In our study, we observed distinct downregulation of FoxO signaling pathway both at the system-level blood transcriptomics and during productive infection, including the FoxO transcription factors FoxO1 and FoxO3, that can act as negative feedback regulators of the innate cellular antiviral response (*Lei et al., 2013*). FoxO1 and FoxO3 also play an essential role in the immunometabolic dynamics and are important targets for glycolysis and gluconeogenesis (*Lundell et al., 2019*).

One of the key pathways that were significantly upregulated both in patients' transcriptomics and during progressive infection in Huh7 cells was OXPHOS. This indicates that CCHFV may manipulate mitochondrial dynamics for its replication by activating the OXPHOS machinery to meet elevated energy demands. Several RNA viruses like respiratory syncytial viruses (RSV), HCV, DENV, Zika virus (ZIKV), and pathogenic human coronaviruses, and are known to target mitochondria for their replication (*Gatti et al., 2020*). Our data also showed that upon suppression of the glycolysis and glutaminolysis that fuels mitochondrial OXPHOS, there was inhibition of CCHFV-replication, further supporting the role of mitochondrial metabolism and biogenesis in CCHFV-replication and pathogenesis. Further investigations on role of mitochondrial biogenesis on CCHFV-pathogenesis can aid novel antiviral strategies.

A shift in OXPHOS can also affect the T-cells differentiation, as observed in both patients' transcriptomics and Huh7 proteomics data. In addition to innate immune responses, adaptive immune responses mediated mainly by T-cells play a critical role in the pathogenesis of viral infections. While we have observed an upregulation of IFN-related pathways in proteomics, there was a downregulation of genes belonging to Th1 and Th2 differentiation and T-cell receptor signaling pathways in the proteomics and the transcriptomics data. Th1 cells secrete IFN-γ, IL-2, TNF-α and are responsible for cell-mediated inflammatory reaction and tissue injury. Th2 cells secrete some of the cytokines including IL-10 and help B-cells for antibody production. During most acute viral infections, there is a cross-regulation for Th1 and Th2 activations primarily mediated by IL-10 and IFN-γ, respectively. Furthermore, activation of Th1 response tends to recovery from an infection while a Th2 activation results a severe clinical pathology (*Mosmann and Sad, 1996*). Th1 and Th2 harmonize the cell-mediated and the humoral response respectively and Th1/Th2 balance has been linked to the prognosis of viral diseases (*Gil-Etayo et al., 2021*). Dengue hemorrhagic fever (DHF), a severe form of dengue fever (DF), is characterized by shock, hemorrhage, and death. It was shown a shift from the predominance of Th1-type response in cases of DF to the Th2-type in cases of DHF (*Chaturvedi et al., 1999*). Mouse model studies have shown activation of the Th1 response is associated with better protection to CCHFV-infection (*Hawman et al., 2021*; *Hinkula et al., 2017*). While activation of Th2 is often associated with disease severity in viral hemorrhagic fever (*Sancakdar et al., 2014*), in case of

CCHFV, balanced Th2-response was shown to be protective in immunized mice with a dynamic shift from Th1 to Th2 at the later part of infection (*Hinkula et al., 2017*). Our patient data and cell infection data suggest that the virus subverts this adaptive immune response by suppressing T-cell response that could influence the disease outcome and recovery. However, this suppression did not impact patient survival in our cohort through Th2 cytokines IL8 and IL10 were significantly elevated in the serum of severe cases. Our data indicated a down-regulation of Th1 and Th2 cell differentiation during acute phase of infection and at the early phase of viral replication. The naive T cells are dependent on OXPHOS while activated T-cells on glycolysis and after differentiation, the cells are mainly dependent upon the glycolysis than OXPHOS (*Angajala et al., 2018*). Switch in the OXPHOS during the CCHFV-infection and imbalance in Th1, Th2, and Th17 differentiation can alter the outcome of the adaptive immune response in survived CCHFV-infected patients.

One of the primary antiviral defense mechanisms is the type-I interferon (IFN-I) response. IFN-I are pleiotropic cytokines with varied cellular functions mediated by the transcriptional activations of several interferon-stimulated genes (ISGs). It is known that CCHFV-replication is sensitive to IFN-I (*Andersson et al., 2006*). However, the virus can also delay the induction of IFN, and IFN treatment is ineffective following the establishment of infection, suggesting that CCHFV has developed mechanisms to block innate immune responses (*Andersson et al., 2008*). The protective role of IFN-I against CCHFV has been exemplified in animal models in which IFNAR$^{-/-}$ or STAT-1$^{-/-}$ mice (*Bente et al., 2010*; *Zivcec et al., 2013*) or STAT2$^{-/-}$ hamsters (*Ranadheera et al., 2020*) showed enhanced susceptibility to CCHFV-infection. Even in in vitro experiments, pre-treatment of cells with IFN-α was found to be inhibitory to CCHFV (*Andersson et al., 2008*). Although CCHFV is inhibited by the IFN-response, not many ISGs with anti-CCHFV activity have been identified apart from MxA, although ISG20 and PKR have been proposed (*Andersson et al., 2004*) to have anti-CCHFV activity. In the present study, we observed that several ISGs with known or proposed anti-CCHFV activity, *i.e.*, Mx1, ISG15 and ISG20 or not defined for CCHFV like IFIT1, IFIT3, IFITM3, IFI16, and OAS3 were upregulated in the acute phase CCHFV patient samples as well as in the cell-infection model.

Our CCHFV infected Huh7 proteomics data is further strengthened by a recent transcriptomics study performed in CCHFV infected Huh-7 and HepG2 showing significant alterations in IFN-response and upregulation of IFIT1, Mx1, ISG15, IF16 genes in CCHFV-infected Huh7 cells (*Kozak et al., 2020*) as was observed in our proteomics data. The CCHFV-induced ISGs, either alone or in combination with other ISGs can possess specific antiviral activities and regulate IFN-signaling (*Mesev et al., 2019*). The changes in the protein abundance of several ISGs at 24hpi and 48hpi also suggest that they have a dynamic activity during different phases of the virus infection. Furthermore, CCHFV has also evolved mechanisms to evade the immune response through the proteins they express and modifications in the genome (*Guo et al., 2012*; *Scholte et al., 2017*).

In conclusion, our study comprehensively describes the host-immune response against CCHFV that can explain viral pathogenesis. The interplay of the metabolic reprogramming toward the central carbon and energy metabolism and its negative association with biological signaling pathways like Notch/FoxO and PI3K/mTOR/HIF-1 and the IFN-mediated host antiviral mechanism could provide attractive options for therapeutic intervention of CCHF. Further studies on the role of mitochondrial biogenesis and dynamics in CCHFV-infection, replication, and pathogenesis will enhance our understanding of host-virus interactions, leading to the development of new antiviral strategies. Moreover, targeting the central carbon and energy metabolism and components of OXPHOS can be an attractive host-directed therapy during the acute CCHFV-infection by increasing the host antiviral response.

## Materials and methods
### Study design, patients, and sample collection

We enrolled 18 adult patients ( ≥ 18 years) diagnosed with CCHF who were followed up by the clinical service of Infectious Diseases and Clinical Microbiology of Sivas Cumhuriyet University Hospital, Sivas, Turkey. The CCHF patients were divided into three groups using the SGS scores of 1, 2, and 3 (*Bakir et al., 2012*). Blood samples were collected on the admission day (acute stage) and from the survivors, 1 year after their recovery (*Table 1*) following confirmed positive real-time RT-PCR test (Altona Diagnostics, Hamburg, Germany) and/or serology by IgM indirect immunofluorescence antibody (IFA) assay (Euroimmun, Luebeck, Germany). Serum cytokine profiling targeting 22 cytokines/chemokines

was performed by Public Health England using a 22 -plex customized Luminex kit (Merck Millipore, Darmstadt, Germany).

## Cells and viruses

The CCHFV strain IbAr10200 (isolated initially from *Hyalomma excavatum* ticks from Sokoto, Nigeria, in 1966) was used in this study. The small cell carcinoma in the adrenal cortex cells, SW13-ATCC-CCL-105, and human hepatocyte-derived cellular carcinoma cell line Huh7 was obtained from Marburg Virology Laboratory (Philipps-Universität Marburg, Marburg, Germany) and matched the STR reference profile of Huh7. The cell lines were tested negative for mycoplasma contamination.

## RNA sequencing (RNAseq) analysis

Peripheral blood mononuclear cells (PBMCs) RNA sequencing (RNAseq) from acute phase and convalescent phase of CCHFV-infected patients was performed as described by us recently (*Appelberg et al., 2020*; *Zhang et al., 2018a*). Total RNA was extracted from Trizol-treated PBMC using the Direct-zol RNA Miniprep (Zymo Research, CA, USA) according to the manufacturer's protocol. RNA-Seq was performed at the National Genomics Infrastructure, Science for Life Laboratory, Stockholm, Sweden, as described by us previously (*Zhang et al., 2018a*). The transcriptomics data pre-processing, alignment, and read counting were performed as described by us recently (*Appelberg et al., 2020*).

All the downstream analysis was performed only on protein-coding genes. Firstly, sample similarity and dissimilarity were accessed through dimensionality reduction using Uniform Manifold Approximation and Projection (UMAP). Normalized expression data of all protein-coding genes were subjected to UMAP dimensionality reduction using R package UMAPv0.2.6.0. The reduced dimensions of the data were plotted in 2D space using the R package ggplot2 v3.3.2 (https://cran.r-project.org/web/packages/ggplot2/index.html). Differential gene expression analysis was performed using raw read counts using the R/Bioconductor package DESeq2 v1.26.0 (*Love et al., 2014*). Genes with adjusted p-values < 0.05 were considered significantly regulated. Further, functional enrichment analysis was done on differential gene expression analysis results to identify significantly regulated pathways. The analysis was carried out using R package PIANO v2.2.0 (*Väremo et al., 2013*) (nperm = 500, geneset stat = mean). Nominal p-values and log2 fold change values of all genes are inputted to the package. Pathways belonging to KEGG category of metabolism, environmental information processing and organismal systems were used as gene-sets for the analysis. Pathways with adjusted p-value < 0.05 were chosen as significantly regulated. Additionally, three gene sets related to IFN-signaling curated by the group (*Chen et al., 2021*) were also considered for the enrichment analysis of gene communities. Gene ontology (GO) enrichment analysis was performed using the enrichr for GO biological process 2018 gene-set (https://maayanlab.cloud/Enrichr/). Redundant GO terms were removed using the online tool REVIGO (*Supek et al., 2011*). Reporter metabolites (*Patil and Nielsen, 2005*) were identified through R package PIANO (nperm = 500, geneset stat = reporter).The human reference genome-scale metabolic model obtained from metabolic atlas (*Robinson et al., 2020*) was used to generate the metabolite-gene sets. Metabolites with adjusted p-values < 0.1 were chosen as significantly regulated (*Radic Shechter et al., 2021*). Metabolic subsystems associated with significant reporter metabolites were extracted from reference metabolic model using in-house Perl scripts. Digital cell type quantification was performed using Estimating the Proportions of Immune and Cancer cells (EPIC) (*Racle and Gfeller, 2020*) algorithm for blood circulating immune cells. Mann-Whitney U test was performed to identify significantly changed cell types.

### Network analysis

The co-expression network analysis was performed as described previously (*Arif et al., 2021*) and adapted in viral diseases (*Appelberg et al., 2020*; *Mikaeloff et al., 2022*). Networks were built by computing pairwise Spearman rank correlations between all genes after removal of non-expressed (row median FPKM <1) or lowly variant (row variance <0.1) genes and analyzed in igraph for those displaying statistically significant (adjusted p < 0.001) positive correlations. Centrality analysis was performed by computing degree centrality. Communities were identified by modularity maximization through the Leiden algorithm (*Traag et al., 2019*). Functional enrichment analysis of network communities was carried out using enrichr module of python package GSEAPY v0.9.16 (*Subramanian et al., 2005*; *Chen et al., 2013*) (https://github.com/zqfang/GSEApy).

## Visualization

Heatmaps were generated using the R/Bioconductor package ComplexHeatmapv2.2.0 (*Gu et al., 2016*) Bubble plots, MA plots, volcano plots, violin plots and bar plots were created using the R package ggplot2 v3.3.2. Network visualization was performed using Cytoscape v3.6.1 (https://cytos-cape.org/). Venn diagrams were constructed using the online tool InteractiVenn (http://www.interac-tivenn.net/).

## In vitro infection assays in Huh7 and SW13 cells

Huh7 and SW13 cells were infected with the CCHFV in triplicate, as described by us previously (*Appel-berg et al., 2020*; *Krishnan et al., 2021*). Briefly, Huh7 cells were infected with CCHFV IbAr10200 at a multiplicity of infection (MOI) of 1. After 1 hr of incubation (37 °C, 5% $CO_2$) the inoculum was removed, the cells were washed with PBS, and 2 ml DMEM supplemented with 5% heat-inactivated FBS was added to each well. Samples were collected in triplicate at 24 and 48hpi along with controls. Due to high permissiveness, we restricted the SW13 infection of 1 MOI for 24 hr only. The infection in Huh7 24hpi was confirmed by immunofluorescence staining of CCHFV nucleoprotein-protein. The cells were fixed in ice-cold acetone-methanol (1:1) and stained using a rabbit polyclonal anti-CCHFV nucleocapsid antibody (home-made) followed by a fluorescein isothiocyanate (FITC)-conjugated anti-rabbit antibody (Thermo Fisher Scientific, US) and DAPI (Roche, US).

## Tandem mass tag (TMTpro) labeled reversed-phase liquid chromatography mass-spectrometric (RPLC-MS/MS) analysis

The RPLC-MS/MS of the TMTpro labeled samples was performed as described by us recently (*Appel-berg et al., 2020*; *Chen et al., 2021*). Briefly, following the protein digestion in S-Trap microcolumns (Protifi, Huntington, NY), the resulting peptides were labeled with TMTpro tags. Labeled peptides were fractionated by high pH (HpH) reversed-phase chromatography, and each fraction was analyzed on an Ultimate 3,000 UHPLC (Thermo Scientific, San Jose, CA) in a 120 min linear gradient. Proteins were searched against the SwissProt human database and CCHFV strain Nigeria/IbAr10200/1970 separately using the search engine Mascot v2.5.1 (MatrixScience Ltd, UK) in Proteome Discoverer v2.5 (Thermo Scientific, US) software allowing up to two missed cleavages.

## Proteomics data analysis

The raw data were first filtered to remove missing data. Proteins detected in all samples were retained for analysis resulting in 8501 proteins in the filtered dataset. The filtered data were then normalized by applying eight different methods using R/Bioconductor package NormalyzerDE v1.4.0 (http://quantitativeproteomics.org/normalyzerde). The quantile normalization was superior to other methods and was selected for further use. Differential protein expression analysis was performed using R/Bioconductor package limma v3.42.2 (https://bioconductor.org/packages/release/bioc/html/limma.html). Proteins with adjusted p-values of less than 0.05 were regarded as significant. KEGG pathway enrichment analysis was performed as mentioned in the transcriptomics section. The mass spectrometry proteomics data have been deposited to the ProteomeXchange Consortium via the PRIDE partner repository with the dataset identifier PXD022672.

## RNAscope and western blot

The *RNAscope* ISH Assays (ACD Bioscience, US) targeting IFI27 (440111, ACD Bioscience, US) and CCHFV (510621, ACD Bioscience, US) were performed as described previously ( *Zhang et al., 2018b*). The western blot (WB) analysis targeting RIG-I, IFIT1, ISG20, ISG15, MX1, and MX2 were performed as described by us previously (*Chen et al., 2021*).

## Metabolic perturbation and virus infection

To inhibit glycolysis and glutaminolysis, following 1hpi (moi 0.1) the cells were treated with 2-deoxy-D-glucose (2-DG, 5 mM), and diazo-5-oxo-L-norleucine (DON, 50 uM) respectively. The concentrations were selected based on the minimal [mean (SD) cell viability, DON-SW13: 84% (4%), DON-Huh7: 78% (2%) and 2-DG-SW13: 80% (2%) or no cytotoxicity (2-DG in Huh7]) in the respective cells 24 hr following drug treatment. The cells were collected after 24hpi and the cells were lysed in Trizol reagent. RNA was extracted using the Direct-zol RNA Miniprep kit (Zymo Research, Irvine,

CA) according to the manufacturer's instructions. Viral RNA was measured by quantitative real real-time polymerase chain reaction (qRT-PCR) using TaqMan Fast Virus 1-Step Master Mix (Thermo Fisher Scientific) with primers and probe specific for the CCHFV L gene; Forward: 5-GCCAACTGTGACKGTK TTCTAYATGCT-3', Reverse-1: 5'- CGGAAAGCCTATAAAACCTACCTTC-3', Reverse-2: 5'-CGGAAAGC CTATAAAACCTGCCYTC-3' and Reverse-3: 5'-CGGAAAGCCTAAAAAATCTGCCTTC-3' and Probe FAM-CTGACAAGYTCAGCAAC –MGB. RNAse was used as endogenous control. The cycling reactions were performed using a capillary Roche LightCycler 2.0 system.

## Acknowledgements

The study is funded by Swedish Research Council Grants 2018–05766 and 2017–03126 to AM and 2017–01330 and 2021–00993 to UN, PHE Grant In Aid 109,509 (inc.inc Ph.D. studentship [EK]) and EU-H2020 CCHFVaccine to RH. The plasma and PBMC sample processing part were performed at Sivas Cumhuriyet University Advanced Technology Application and Research Center (CUTAM), Sivas, Turkey, supported by EU-H2020 CCHFVaccine. The authors would like to acknowledge support from the National Genomics Infrastructure (NGI), Science for Life Laboratory, for RNAseq and Proteomics Biomedicum, Karolinska Institute, Solna, for LC-MS/MS analysis. The authors would like to acknowledge Dr. Shubha Krishnan for the critical reading of the manuscript and comment, Maike Sperk, Xi Chen for assisting in some laboratory experiments. The computations were performed using resources provided by SNIC through the Uppsala Multidisciplinary Center for Advanced Computational Science (UPPMAX) under Project SNIC2017-550. The microscopy part of the study was performed at the Live Cell Imaging Facility and Biomedicum Imaging Core, Karolinska Institute, Sweden, supported by grants from the Knut and Alice Wallenberg Foundation, the Swedish Research Council, the Centre for Innovative Medicine, and the Jonasson Center at the Royal Institute of Technology, Sweden.

## Additional information

### Funding

| Funder | Grant reference number | Author |
|---|---|---|
| Vetenskapsrådet | 2017-01330 | Ujjwal Neogi |
| Vetenskapsrådet | 2021-00993 | Ujjwal Neogi |
| Vetenskapsrådet | 2018-05766 | Ali Mirazimi |
| Vetenskapsrådet | 2017-03126 | Ali Mirazimi |
| European Commission | 732732 | Roger Hewson |

The funders had no role in study design, data collection and interpretation, or the decision to submit the work for publication.

### Author contributions
Ujjwal Neogi, Conceptualization, Methodology, Project administration, Resources, Supervision, Visualization, Writing – original draft; Nazif Elaldi, Data curation, Project administration, Resources, Writing – review and editing, Investigation; Sofia Appelberg, Emma Kennedy, Soham Gupta, Sara Svensson-Akusjärvi, Vanessa Monteil, Formal analysis, Methodology, Writing – review and editing; Anoop Ambikan, Data curation, Formal analysis, Methodology, Software, Writing – review and editing; Stuart Dowall, Binnur K Bagci, Formal analysis, Investigation, Writing – review and editing; Jimmy E Rodriguez, Data curation, Formal analysis, Methodology, Writing – review and editing; Akos Vegvari, Data curation, Formal analysis, Methodology, Supervision, Writing – review and editing; Rui Benfeitas, Data curation, Formal analysis, Software, Supervision, Validation, Writing – review and editing; Akhil Banerjea, Investigation, Resources, Writing – review and editing; Friedemann Weber, Data curation, Funding acquisition, Project administration, Resources, Supervision, Writing – review and editing; Roger Hewson, Data curation, Funding acquisition, Methodology, Project administration, Resources, Supervision, Writing – review and editing; Ali Mirazimi, Conceptualization, Funding acquisition, Project administration, Resources, Writing – review and editing

## Author ORCIDs

Ujjwal Neogi http://orcid.org/0000-0002-0844-3338
Nazif Elaldi http://orcid.org/0000-0002-9515-770X
Anoop Ambikan http://orcid.org/0000-0002-5221-9085
Vanessa Monteil http://orcid.org/0000-0002-2652-5695
Rui Benfeitas http://orcid.org/0000-0001-7972-0083

## Ethics

Human subjects: This study was approved by the Local Research Ethics Committee of the Ankara Numune Education and Research Hospital, Turkey (Protocol # 17-1338) and Regional Ethics Committee, Stockholm (Dnr. 2017-/1712-31/2). All patients and/or their relatives were informed about the purpose of the study and signed a consent form before collection.

## Decision letter and Author response

Decision letter https://doi.org/10.7554/eLife.76071.sa1
Author response https://doi.org/10.7554/eLife.76071.sa2

---

# Additional files

## Supplementary files

• Supplementary file 1. The DGE profile for the acute phase compared to the recovered phase in all patients.
• Supplementary file 2. Pathways significantly regulated by genes expressed at the acute infection phase compared to the recovered phase identified in PIANO.
• Supplementary file 3. Pathways significantly regulated by proteins in mock and CCHFV-treated Huh7 cells following 24hpi and 48hpi and time-series analysis identified in PIANO.
• Transparent reporting form
• Source data 1. Raw western blot images.

## Data availability

All data needed to evaluate the conclusions in the paper are present in the paper and/or the Supplementary Materials. Raw RNAseq data is avaible in Sequence Read Archive (SRA) with PRJNA680886. The mass spectrometry proteomics data have been deposited to the ProteomeXchange Consortium via the PRIDE partner repository with the dataset identifier PXD022672. All the codes are available in GitHub (https://github.com/neogilab/CCHF-Turkey; copy archived at swh:1:rev:e8b869e22f4133857174e7a04cb1994ed3467a32). The raw western blot images are provided as source data 1.

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
