## [Editor Report]

The data presented here provide novel insight into the host response to CCHFV infection. These data further our understanding of how CCHFV causes disease in humans and will support the development of therapeutics to address the significant morbidity and mortality caused by this virus.

---

## [Decision Letter]

**Decision letter after peer review:**

Thank you for submitting your article "Multi-omics insights into host-viral response and pathogenesis in Crimean-Congo Hemorrhagic Fever Viruses for novel therapeutic target" for consideration by *eLife*. Your article has been reviewed by 2 peer reviewers, one of whom is a member of our Board of Reviewing Editors, and the evaluation has been overseen by Betty Diamond as the Senior Editor. The reviewers have opted to remain anonymous.

Essential revisions:

While reviewer 1 has mostly minor comments, reviewer 2 has several suggestions/requests for improving data analysis and conclusions. In particular, authors should address comments 2 and 4 from reviewer 2 along with providing more context on the pathways identified and any potential co-variates identified in the patients.

*Reviewer #1:*

In the manuscript by Neogi et al., the authors report data on RNAseq of PBMCs from CCHFV infected humans and proteomics analysis on CCHFV-infected cells. Interestingly, authors found that CCHFV-infection caused perturbations to metabolic pathways and also upregulated ISGs, consistent with a viral infection. Importantly, authors discuss their findings in the context of other viral infections to help readers understand how these pathways may influence CCHF pathogenesis. The use of longitudinal samples during acute disease and well after recovery is helpful to account for variability in human populations. These findings are significant as the human host-response to CCHFV is understudied and findings such as those presented here will help researchers using animal models determine the similarities and differences in these systems to human CCHF cases. The manuscript is well written and the extensive discussion helpful to understand the data. A limitation is their cohort had only one patient in the most severe disease group making conclusions on mechanisms that may mediate poor outcome difficult. Cumulatively, these findings add to our knowledge about how the human host responds to CCHFV infection, an area that warrants continued research.

Line 138-140 and 363-368: Do authors have data, e.g. ELISpot data, to show that patients did or did not develop T-cell responses to CCHFV?

Figure 4G: Do authors have cell viability data? Is decrease in CCHFV replication due to general decrease in cell viability?

Line 353 – 354: Authors should clarify that this statement is not based on CCHFV-specific data.

Line 211-212: Why was the extensively mouse passaged 10200 strain used? Would it not make more sense to use a Turkish CCHFV strain?

Line 214: An MOI of 1 via a poisson distribution would infect 63% of cells which would leave few uninfected cells for a second round of infection and little to no cells for a third round.

Line 467: In our experience, SW13s take 3 – 4 days to show cytopathic effect upon infection. Did authors observe CPE by 48 HPI?

Figure 1B: I would recommend authors use colors other than red and green to make figure more accessible to color-blind readers.

Line 403 – 405: What is the feasibility of targeting host metabolism as a host-directed therapy? Have therapies along these lines been evaluated pre-clinically or clinically for viral infections? Would it not be expected that targeting this pathway, an essential component to cell viability, would result in substantial negative effects on the host?

*Reviewer #2:*

The manuscript by Neogi, U. et al., provides analysis of the circulating immune responses to CCHFV infections in patients. To my knowledge, this is a truly unique dataset as it represents one of the first analysis of the circulating response in patient samples. The authors then further characterize the immune response using in vitro proteomics to isolate pathways associated with glycolysis that could serve as therapeutic targets and provide preliminary evidence that these pathways are important for the virus to replicate in cell lines. Some of the strengths of the manuscript are providing a resource for other to investigate host responses to CCHFV, detailed information on patient samples for further investigation into factors such as sex and age on the host response to CCHFV infections and provides a proof of concept for utilization of the information for potential therapeutics. The paper does rely too heavily on methods, such as WGCNA, whose results can be easily biased in datasets with high variance and low n, such as with patient data. Additionally, more context on the pathways provided beyond their identification is needed. Finally, assessment on how any potential co-variates identified in the patients may bias the results is needed to isolate pathways altered by the host response to the infection.

Specific recommendations that would strengthen the manuscript are listed below.

1. In the Results section on sample collection and clinical data, it would be useful to have a quick note about the clinical severity scores, how they are calculated (is it a standard sheet, is it the same individual scoring all patients, are there any potential biases in the scoring?). It would also be useful to have a visual of the distribution of the clinical severity scores to help put them into context for the reader. Especially since they are utilized so heavily, there was just not a lot of context for these values.

2. The authors use a whole multitude of different p-value cutoffs for significance. There should be a justification for each cutoff being used. For example, on line 144 the authors state that for the metabolite profiling, a adj. p < 0.1 was used but for the DGE an adjust p-value of 0.05 was used. Have the authors done prior work to determine the optimal cutoffs for each of these analysises? If so, that should be included. If not, there needs to be a justification for each set of p-values analyzed.

3. The IFN signature is not surprising and is a common feature in patients that present with severe viral infections. Additionally, this was predicted from NHP data (see Arnold, CE. et al., Scientific Reports 2021). However, the authors could provide a greater context to the IFN and ISG signatures. What percent of the ISGs are attributed to Type I vs. Type II IFN? This is especially important given the NK/Th17 type responses that are detected.

4. WGCNA analysis should always be done with caution when using samples that have high variance (such as patient data) and/or a now n (only 12 samples in this case). Additionally, there is no assessment on co-variate assessment and how factors such as age, sex, clinical score, days since symptom onset, ect. could have affected these results. This is particularly important in-patient data where all of these variables cannot be controlled.

5. For the proteomics data from the cell lines, it would be good to also know what the clustering looks like with the virus removed to help prevent biases caused by differing viral loads.

6. Additionally for the proteomics data, have the authors done an assessment of the viral load? Based on the IF, they are potentially past the exponential growth phase. If this is the case, data collected during the exponential growth phase would be useful to know what pathways are needed for early replication. Growth curve analysis would also help put this into context.

7. For all the UMAP analysis, how many features were used in the UMAP calculation and was a preliminary dimensionality reduction used first?

8. From the sequencing data, it would be great if the authors could perform cellular deconvolution on the samples as well to determine if there is likely changes in cell types causing changes in gene expression. This would be helpful for the NK/Th17 type responses detected but also the reduction in HLA genes.

9. How are the other identified genes sets related to viral load. There was an assessment of ISGs to viral load which showed the expected pattern. It would interesting to know which other gene sets correlate strongly with viral load and particularly ones that seem to not be as linked to viral load. Correlation analysis of gene modules would provide much greater context to the identified gene sets.

---

## [Author Response]

Essential revisions:While reviewer 1 has mostly minor comments, reviewer 2 has several suggestions/requests for improving data analysis and conclusions. In particular, authors should address comments 2 and 4 from reviewer 2 along with providing more context on the pathways identified and any potential co-variates identified in the patients.

We are thankful to both the reviewers for their positive response and constructive criticism in our study. We have now addressed all the comments including comment 2 and 4 with additional analysis.

Reviewer #1:Line 138-140 and 363-368: Do authors have data, e.g. ELISpot data, to show that patients did or did not develop T-cell responses to CCHFV?

We are thankful to the reviewer for a very insightful question. It would have been very interesting to look into the T-cell response to CCHFV in the patients. Unfortunately, we are unable to perform ELISpot assay due to a lack of protocols and infrastructure to perform the same in the BSL4.

Figure 4G: Do authors have cell viability data? Is decrease in CCHFV replication due to general decrease in cell viability?

The reviewer is concerned if the observed decrease in CCHFV replication is due to any cytotoxicity-mediated cell death. We would like to assure the reviewer that we have performed cytotoxicity assay and have selected the concentration having cell viability of ≥80%. Thus, the observed inhibition of CCHFV infection is an antiviral effect of the drug. We already mentioned the cell viability in the earlier version:

“The concentrations were selected based on the minimal [mean (SD) cell viability, DON-SW13: 84% (4%), DON-Huh7: 78% (2%) and 2-DG-SW13: 80% (2%) or no cytotoxicity (2-DG in Huh7)] in the respective cells 24hrs following drug treatment.”

Line 353 – 354: Authors should clarify that this statement is not based on CCHFV-specific data.

We are thankful for the suggestion. The original statement “During the acute viral infections, there is a cross-regulation for Th1 and Th2 activations primarily mediated by IL-10 and IFN-γ, respectively.” Is now modified as follows “ During most acute viral infections, there is a crossregulation for Th1 and Th2 activations primarily mediated by IL-10 and IFN-γ, respectively.

Line 211-212: Why was the extensively mouse passaged 10200 strain used? Would it not make more sense to use a Turkish CCHFV strain?

We agree with the reviewer that the Turkish CCHFV strain might have been ideal. The problem we have faced has been to manage to culture this virus in high titer, which was necessary for this set of experiments. Therefore, we used the ref CCHFV-strain 10200 available in the European virus archive and produce high titer viruses.

Line 214: An MOI of 1 via a poisson distribution would infect 63% of cells which would leave few uninfected cells for a second round of infection and little to no cells for a third round.Line 467: In our experience, SW13s take 3 – 4 days to show cytopathic effect upon infection. Did authors observe CPE by 48 HPI?

We do observe a higher amount of cell death and detachment of SW13 cells from the plate at 48 h with moi 1 in comparison to the Mock-infected control as well as 24 hpi.

Figure 1B: I would recommend authors use colors other than red and green to make figure more accessible to color-blind readers.

We thank the reviewer for the suggestion. We have now changed the color red and green to orange and blue.

Line 403 – 405: What is the feasibility of targeting host metabolism as a host-directed therapy? Have therapies along these lines been evaluated pre-clinically or clinically for viral infections? Would it not be expected that targeting this pathway, an essential component to cell viability, would result in substantial negative effects on the host?

This is a very interesting question. We are extensively studying this in SARS-CoV-2 (Krishnan et al., 2021, Molecular and Cellular Proteomics), and HIV-1 (Mikaeloff et al., 2022, *Nature Communication Biology*) apart from the CCHFV, however those are all pre-clinical studies. We are initiating animal experiments with SARS-CoV-2, CCHFV and HIV-1 with these drugs.

2-DG: As of now 2-DG is approved in India for emergency use in SARS-CoV-2 severe patients. However, no data was released. It may not be ideal choice in mild or moderate cases. Rather more for the severe lifesaving cases.

Though no scientific literature available, the press release of the Indian Govt (https://pib.gov.in/PressReleasePage.aspx?PRID=1717007), reported that the Phase 2 trials of the 2-DG arm showed faster symptomatic cure than Standard of Care (SoC) arm on various endpoints (vital parameters) with a significantly favorable trend (2.5 days difference). In the 2-DG arm, 42% of the patients improved symptomatically and became free from supplemental oxygen by Day-3 compared to 31% in the SoC arm, indicating an early relief from oxygen therapy/dependence. A higher proportion of patients treated with 2-DG showed RT-PCR negative conversion in COVID patients.

DON: A recent study in HIV-1 showed that DON reversed cognitive impairment in EcoHIV-infected mice in HIV-associated neurocognitive disorders (HAND).

So, we believe there is high feasibility of targeting host metabolism as a host-directed therapy during severe cases but more research is warranted to take it to the clinical trials.

Reviewer #2 (Recommendations for the authors):Specific recommendations that would strengthen the manuscript are listed below.1. In the Results section on sample collection and clinical data, it would be useful to have a quick note about the clinical severity scores, how they are calculated (is it a standard sheet, is it the same individual scoring all patients, are there any potential biases in the scoring?). It would also be useful to have a visual of the distribution of the clinical severity scores to help put them into context for the reader. Especially since they are utilized so heavily, there was just not a lot of context for these values.

We are thankful to the reviewer for pointing out the lack of clarity in defining the severity grading scores. The individual patients' information has already presented the table 1. We have now given the daily SGS score during the hospitalization as supplementary figure 1. As per reviewer’s suggestion we have now added that in the result section as follows:

“We used the severity grading scores (SGS) to define the CCHF severity that calculated using age, clinical findings (bleeding, hepatomegaly, organ failure), routine laboratory parameters (blood levels of liver enzymes and lactate dehydrogenase, blood platelet and leucocyte counts, blood coagulation tests (prothrombin time, D-dimer and fibrinogen)) (Bakir et al., 2012). By using these criteria, a standard SGS sheet for each patient was filled by the infectious diseases physician on admission day. By using SGS criteria, 33% (6/18) patients were grouped into severity group 1 (SG-1), 61% (11/18) patients into severity group 2 (SG-2) and, 6% (1/18) patients into severity group 3 (SG-3).”

The CCHF patient characteristics are summarized individually in Table 1 and the calculated daily SGS scores during hospitalization in Table 1-source data 1.

2. The authors use a whole multitude of different p-value cutoffs for significance. There should be a justification for each cutoff being used. For example, on line 144 the authors state that for the metabolite profiling, a adj. p < 0.1 was used but for the DGE an adjust p-value of 0.05 was used. Have the authors done prior work to determine the optimal cutoffs for each of these analysises? If so, that should be included. If not, there needs to be a justification for each set of p-values analyzed.

We are thankful for the suggestion. We have sought to balance the number of potential false positives at each analysis, considering the background and number of statistic tests for our sample size. We aimed to use adjusted p value<0.05 as threshold for significance throughout except the reporter metabolite analysis or co-expression analysis (see Q4). In the reporter metabolites, multiple hypothesis correction was performed for 3600 predicted metabolites which is much less than the number of genes analyzed in differential expression analysis (~20,000). By keeping adjusted p-value cut-off of 0.1 will only cause for low number of false positives (10%), and considering the detected 37 statistically significant metabolites, we estimate that <4 reporter metabolites to be potential false positives. Though there were no consensus, adj p<0.1 has been used earlier for reporter metabolite analysis (Ref: Shechter et al., 2021, Mol Syst Biol). For better readability we have now elaborated the methods section with appropriate reference.

3. The IFN signature is not surprising and is a common feature in patients that present with severe viral infections. Additionally, this was predicted from NHP data (see Arnold, CE. et al., Scientific Reports 2021). However, the authors could provide a greater context to the IFN and ISG signatures. What percent of the ISGs are attributed to Type I vs. Type II IFN? This is especially important given the NK/Th17 type responses that are detected.

We agree with the reviewer that changes in the expression of Interferon stimulated genes upon CCHFV infection were expected, as is observed in many other viral infections, However, the induction of different ISG’s differs in different viruses and the knowledge is valuable. The reviewer is true that delineating type-I and Type-II stimulated ISGs can reveal about the regulation of immune cells. Yet, the Interferon signaling is a complex network and majority of the ISGs can be induced by either of the type-I and/or type-II and even type-III IFNs. The overlap of the ISGs in different types of IFN signaling pathways is shown in (Author response image 1) . Thus, in this study we would desist from attributing the ISGs to any specific IFN-response and specifying their context to the regulation of cellular immunity.

**Author response image 1. sa2fig1:** 

4. WGCNA analysis should always be done with caution when using samples that have high variance (such as patient data) and/or a now n (only 12 samples in this case). Additionally, there is no assessment on co-variate assessment and how factors such as age, sex, clinical score, days since symptom onset, ect. could have affected these results. This is particularly important in-patient data where all of these variables cannot be controlled.

We are thankful for the comment. We agree that the low number of samples is a problem for any correlation or regression analysis. We did not use WGCNA. We have our weighted coexpression analysis algorithm for the analysis that we have recently adopted for infected material (Appelberg et al., 2021). We are sorry that we have not explicitly mentioned in the manuscript.

We have now elaborated the method sections with appropriate reference and all the codes are deposited in the GitHub repository. Our study design is longitudinal not comparing two groups (casecontrol). However as suggested we performed association analysis with the age, ct value and SGS score with 5076 genes (after excluding lowly expressed (FPKM <1) and low variant (variance < 0.1) genes). We observed some high correlation coefficients (Spearman’s Rho) but none of the adjusted p values were <0.05. However, this could be interpreted carefully as the sample size is low and high variance. Please see the result: https://figshare.com/s/0b9016e61e8c54c4aa3d

Therefore, we also have considered stringent adjusted p-value cut-off of 0.001 to avoid maximum number of false-positive outcomes and Spearman Rho greater than 0.84. As indicated above (Q2), this cut-off was chosen to balance the number of potential false positives considering the number of statistical tests being performed. At this false discovery rate, we estimate ~323 correlations to be potential false positives. We have now added this information to the methods.

5. For the proteomics data from the cell lines, it would be good to also know what the clustering looks like with the virus removed to help prevent biases caused by differing viral loads.

We would like to assure the reviewer that the clustering (Figure 4A) was performed not including the viral proteins. This is also reflected in Figure 5A heatmap of proteins associated with IFN signaling. We are sorry that we have not mentioned that earlier. We have now added in the method section as follows:

“Proteins were searched against the SwissProt human database and CCHFV strain Nigeria/IbAr10200/1970 separately using the search engine Mascot v2.5.1 (MatrixScience Ltd, UK) in Proteome Discoverer v2.5 (Thermo Scientific, US) software allowing up to two missed cleavages.”

“Figure 4. LC-MS/MS based quantitative proteomics analysis in CCHFV-infected Huh7 and SW13 cells. (A) Principal component analysis of proteomics samples of Huh7 cells and SW13 (inset) using only human proteins.”

6. Additionally for the proteomics data, have the authors done an assessment of the viral load? Based on the IF, they are potentially past the exponential growth phase. If this is the case, data collected during the exponential growth phase would be useful to know what pathways are needed for early replication. Growth curve analysis would also help put this into context.

We are thankful for the comment. We have not performed the viral load for the proteomics analysis. We do agree that based on the IF, they are potentially past the exponential growth phase. This is because we want the majority of the cells to be infected while doing proteomics analysis to avoid the bystander effect. Additionally, as shown in Figure 4D there were difference in regulations of the pathways at 24hpi and 48hpi in huh7 cells which indicates that the virus regulates different pathways during different phases of infection.

7. For all the UMAP analysis, how many features were used in the UMAP calculation and was a preliminary dimensionality reduction used first?

Normalized expression values of all the protein-coding genes were used for UMAP dimensionality reduction. While it is desirable to employ a preliminary dimensionality reduction (e.g. PCA) for speeding computations of neighboring distances, this was not necessary and was a step that we did not have to employ.

8. From the sequencing data, it would be great if the authors could perform cellular deconvolution on the samples as well to determine if there is likely changes in cell types causing changes in gene expression. This would be helpful for the NK/Th17 type responses detected but also the reduction in HLA genes.

We are thankful to the reviewer for the suggestion. We have performed the cellular deconvolution using the EPIC algorithm. We didn’t see any statistically significant difference between the cell types predicted. We have now added this in the manuscript as follows:

“To check whether the gene expression changes between the acute phase and recovered phase may be due to differences in cell types abundances, we performed digital cell quantification (DCQ) using the Estimating the Proportions of Immune and Cancer cells (EPIC) (Racle and Gfeller, 2020) algorithm for blood circulating immune cells. No statistically significant (adj p<0.05) difference was observed in the key immune cell types (Figure 1—figure supplement 1).”

9. How are the other identified genes sets related to viral load. There was an assessment of ISGs to viral load which showed the expected pattern. It would interesting to know which other gene sets correlate strongly with viral load and particularly ones that seem to not be as linked to viral load. Correlation analysis of gene modules would provide much greater context to the identified gene sets.

We agree with the reviewer and as per the reviewer’s suggestion, we have performed association analysis with age, CT value, and SGS score with 5076 genes (after excluding lowly expressed (FPKM <1) and low variant (variance <0.1) genes). We observed some high correlation coefficients (Spearman Rho) but none of the adjusted p values were <0.05. However, this requires cautious interpretation as the sample size is low and variance is high. Given that we don’t have the functional data we have been restricted from commenting upon that. But it will be a great avenue to research further.

Please see the analysis: https://figshare.com/s/0b9016e61e8c54c4aa3d.